# Optimization of Structural Parameters and Mechanical Performance Analysis of a Novel Redundant Actuation Rehabilitation Training Robot

**DOI:** 10.3390/biomimetics10040199

**Published:** 2025-03-25

**Authors:** Junyu Wu, He Wang, Yubin Liu, Zhuoqi Man, Xiaofan Yang, Xuanming Cao, Hegao Cai, Jie Zhao

**Affiliations:** 1State Key Laboratory of Robot Technology and Systems, Harbin Institute of Technology, Harbin 150001, China; 21B908013@stu.hit.edu.cn (J.W.); 23B308002@stu.hit.edu.cn (Z.M.); 22B908042@stu.hit.edu.cn (X.Y.); 23B908038@stu.hit.edu.cn (X.C.); hgcai@hope.hit.edu.cn (H.C.); jzhao@hit.edu.cn (J.Z.); 2College of Intelligent Systems Science and Engineering, Harbin Engineering University, Harbin 150001, China; wang_he@hrbeu.edu.cn

**Keywords:** dynamic simulation, performance analysis, rehabilitation training robot, redundant structure, structure parameter optimization

## Abstract

The integration of redundant structures into robotic systems enhances the degrees of freedom (DOFs), flexibility, and capability to perform complex tasks. This study evaluates the mechanical performance of a 9-DOF series-parallel hybrid redundant device designed for rehabilitation training of patients with balance disorders. The redundant structural design improves the robot’s movement flexibility, optimizes load distribution, and mitigates stress concentration in local joints or components. To optimize the robot’s overall structural parameters and reduce joint driving forces, a genetic algorithm (GA) was employed. A custom dataset was created by collecting motion-related data, including foot posture and position. The robot’s mechanical characteristics were comprehensively analyzed, followed by simulation experiments. The results demonstrate that incorporating the redundant structure, along with the optimization of structural parameters, significantly enhances the robot’s mechanical performance. This study provides a solid foundation for the functional development and control system design of rehabilitation robots, extending the capabilities of existing systems and offering a novel, reliable, and efficient therapeutic tool for patients with balance disorders.

## 1. Introduction

Individuals with balance disorders face challenges in maintaining physical stability and coordinating their movements. As the number of affected patients continues to increase, addressing their treatment has become an urgent priority. Exercise training remains the most direct and effective method for managing balance disorders; however, traditional physical therapy is limited by scarce medical resources and individual patient variability. In this context, rehabilitation training robots have emerged as a promising solution to overcome these challenges. These robots provide personalized, efficient, and safe rehabilitation training, reducing the burden on medical professionals and driving technological advancements in the field of rehabilitation medicine [1,2,3].

Rehabilitation training robots are primarily categorized into exoskeleton robots and end-effector robots. Exoskeleton robots are externally mounted on the user’s limbs, providing assistance during movement or training. In contrast, end-effector robots typically incorporate treadmill conveyors or motion pedals that facilitate user movement during rehabilitation [4,5]. Figure 1 illustrates two typical types of rehabilitation robots. The mechanical properties of rehabilitation robots are critical to the effectiveness of treatments, ensuring user safety and improving treatment efficiency. These robots must provide sufficient mechanical support to assist patients in performing various movements while also offering appropriate resistance to promote muscle strength and coordination recovery. Strong mechanical properties are essential for safeguarding patients during training, minimizing the risk of falls or other injuries. Furthermore, rehabilitation robots are capable of simulating natural human movement patterns and applying auxiliary forces to enhance rehabilitation efficiency, thereby accelerating the recovery process [6,7,8]. The 9-DOF redundant robot examined in this study is designed to rehabilitate lower limbs and is particularly suitable for patients with severe balance disorders. During rehabilitation sessions, users stand on exercise pedals and move in accordance with the pedal motion, which imposes high demands on the robot’s load-bearing capacity and mechanical characteristics [9].

Redundant robots have more DOFs than is necessary to complete specific tasks. This redundant structure can considerably improve mechanical performance as well as optimize the joint driving force and energy consumption. Redundant DOFs increase the number of available motion paths and thereby, the flexibility and obstacle avoidance ability. A redundant structure can increase the workspace and operability of the robot, and it can improve the kinematic characteristics of a robot, for example, by preventing singular points. However, redundant structures can also make a robot bulky and complicate control, creating problems such as multiple kinematic solutions [10,11,12]. Figure 2 illustrates the advantages and disadvantages of redundant structures and the importance of the mechanical properties of robots in rehabilitation training.

The mechanical performance analysis of robots refers to the study of the dynamic metrics such as the speed, acceleration, and torque of robots under different operating modes, in order to evaluate and optimize their performance. This analysis is key to ensuring that a robot can accurately and efficiently complete predetermined tasks [13,14,15]. This dynamic study includes dynamic modeling, simulation, and experimental verification. Before a robot is actually manufactured, simulation software can be used to analyze the robot’s mechanical performance in a simulation environment by considering the kinematics, dynamic characteristics, and performance of the robot under different control strategies. Dynamic experiments are physical tests for evaluating the actual performance of robots and identifying potential problems that could be encountered in practical operations, such as mechanical wear and tear [16,17,18,19].

Existing rehabilitation robots, particularly end-effector-based systems, face limitations in mechanical performance and workspace, restricting their rehabilitation motion modes. This study introduces a novel serial-parallel hybrid redundant-driven rehabilitation robot, optimized through structural configuration and parameter adjustments to improve load distribution and motion characteristics. This approach expands the functional capabilities of rehabilitation robots, offering a reliable and efficient solution for treating balance disorders. The robot’s pedal guides patient movements, fostering sensory integration and the active processing of movement information to enhance lower limb and pelvis control. With high load-bearing demands, the robot’s optimized structure supports various rehabilitation modes, such as walking on flat surfaces and stair climbing, ensuring safety, stability, and functional versatility. These improvements lay a solid foundation for future advancements in rehabilitation robot functions and control system design.

This study is organized as follows: 1—introduction of the 9-DOF robot; 2—optimization of the robot’s structural parameters; 3—dynamic simulation and mechanical performance analysis on the basis of walking gait data; and 4—summary.

## 2. Introduction to the Robot Structure

### 2.1. Robot Structural Composition

End-effector rehabilitation training robots are often configured in parallel to achieve high stiffness and load-bearing capacity. A 6-DOF device, utilizing a parallel configuration, is proposed in this study. To further improve the robot’s performance, a redundant 3-DOF structure is introduced. This 3-DOF device is connected in series with the 6-DOF device, resulting in a 9-DOF series-parallel hybrid robot with redundancy. Figure 3 illustrates the rehabilitation training system consisting of two 9-DOF robots. The two robots are arranged in a mirror configuration and work collaboratively to perform the tasks. The motion pedals at the end of each 3-DOF device support the user’s lower limbs, driving the pedals to assist the user in completing walking or other movements. The parallel configuration ensures the rigidity and load-bearing capacity of the robot, while the redundant mechanism enhances flexibility, workspace, and mechanical performance. The embedded connection between the two devices reduces the height of the motion platform, preventing interference between the robot and the user’s body [9].

The 6-DOF device consists of six spherical joint-connecting rod-spherical joint-prismatic joint motion branches (6-SSP), where the transmission mechanism consists of guide rail sliders, with the linear motion pairs serving as the active pairs. The 3-DOF device consists of three motion branches of a Hooke joint-prismatic joint-spherical joint (3-RPS), and it uses linear electric cylinders as the drive. The structural diagrams of the 6-SSP and 3-RPS devices are shown in Figure 4.

The 6-SSP mechanism, known for its high stiffness and load-bearing capacity, serves as the base, while the compact 3-RPS mechanism extends redundancy, enhancing flexibility and workspace. This design supports a wider range of complex rehabilitation movements. The 6-SSP mechanism utilizes a parallel guide rail and slider system, enabling efficient transmission and high-speed, high-acceleration foot motion simulation. The 3-RPS mechanism facilitates rotational movements, such as toe flexion and dorsiflexion. Together, the mechanisms complement each other, optimizing motion performance through effective DOF distribution. Their embedded connection minimizes human–robot interference, while the low pedal height improves suitability for rehabilitation scenarios, enhancing patient service. The redundant structure allows the robot to reach the same target via multiple paths. By allocating DOFs between the two mechanisms, the robot’s motion is optimized to reduce energy consumption, minimize joint wear, and enhance motion smoothness and accuracy. The 6-SSP uses a guide-rail slider group for high-speed linear motion, while the 3-RPS’s compact design facilitates rotational motion. Proper DOF allocation maximizes the advantages of both devices and optimizes their load distribution [9]. The structural compositions of the 6-SSP and 3-RPS devices are shown in Figure 5.

### 2.2. Principle of DOF Allocation

The coupling relationship between the DOFs of the two devices of this redundant robot makes the kinematic solution non-unique. To achieve a unique kinematic solution and ensure control accuracy, the structural characteristics and motion performance of the two devices (such as task orientation and kinematic optimization) must be considered to reasonably allocate the DOFs [20]. The principle of DOF allocation is shown in Table 1 below.

To expand the workspace of the robot, when the robot needs to move linearly in the *Z*-direction, both the 6-SSP and 3-RPS devices must be used proportionally for the motion stroke. Applying this principle of DOF decomposition and motion allocation guarantees the uniqueness of the kinematic solution.

## 3. Optimization of the Robot Structural Parameters

The structural parameters directly affect the robot mechanical performance. These parameters can be optimized to achieve the best mechanical performance. In Figure 6, the key structural parameters of the robot are shown and the joint numbers and coordinate system are defined. The key mechanistic parameters of the 6-SSP device include the distance between the guide rails *D*, the unequal lengths of the six connecting rods (*l* for the rods 1 and 2 and *L* for the rest), and the geometric dimensions *h* and *d* of the motion platform. The key mechanistic parameters of the 3-RPS device include the initial length *H* of the electric cylinder, the distribution range *R* of the Hooke joint, and the distribution range *r* of the spherical joint.

The overall structural parameters of the robot are optimized to reduce joint driving forces, thereby enhancing mechanical performance. To achieve this, an optimization mathematical model is developed, and the objectives, methods, and design variables for optimization are defined. A Genetic Algorithm (GA) is an efficient optimization technique that seeks the optimal solution by simulating natural selection and evolutionary processes. Its global search capability effectively prevents local optima and demonstrates robust performance in solving complex optimization problems, while the GA excels in many applications, it is not always the optimal choice. Other algorithms, such as Simulated Annealing (SA), may outperform GA in certain situations. In this study, we capitalize on the advantages of the GA to address optimization problems involving multiple variables and complex constraints, specifically for the structural parameter optimization of robots.

The 6-SSP device is associated with five continuous types of design variables that constitute a five-dimensional design space ${\left[l,L,D,d,h\right]}^{T}$. The 3-RPS device has a 3-dimensional design space ${\left[R,r,H\right]}^{T}$. A dynamic model of the robot must be established as the objective function for the optimization to create a computational framework and evaluate the performance. However, the complex structure of the robot results in a highly nonlinear dynamic model that is cumbersome to derive. Therefore, results from a mechanical simulation are used, instead of those calculated using a mechanical model. The  GA is used as the optimization method. The basic principle of the GA is to simulate natural selection and genetic mechanisms during the process of biological evolution and search for the optimal solution through iteration. The GA method is unique in the following ways: a group of candidate solutions is processed simultaneously rather than a single solution; genetic operations, such as crossover and mutation, are used to generate new solutions; the quality of the solution is evaluated based on the fitness function; excellent individuals are selected based on their fitness to enter the next generation [21,22,23].

All six linear pairs ${\left[\mathrm{LP}1,\mathrm{LP}2,\mathrm{LP}3,\mathrm{LP}4,\mathrm{LP}5,\mathrm{LP}6\right]}^{T}$ of the 6-SSP device are active motion pairs with a joint driving force of ${\left[\tau_{1},\tau_{2},\tau_{3},\tau_{4},\tau_{5},\tau_{6}\right]}^{T}$. The optimization objective is to reduce the joint driving force during 6-SSP motion, that is, to continuously reduce, and thereby minimize, the maximum driving force among the six joints of the robot in the simulated motion. The structural characteristics of the 6-SSP device make it more difficult to move the motion platform in the *Z*-direction (by overcoming gravity) than in the other directions, that is, a larger joint driving force is required for motion in the *Z*-direction than the other directions. Therefore, linear motion with uniform acceleration in the *Z*-direction is simulated, and the simulation results replace the theoretical results in the optimization process. This strategy can theoretically greatly improve the mechanical performance of the robot for motion in the *Z*-direction. The optimization objective is given by Formula Equation 1 as follows: (1)$$f_{op}=\min\left\{Z\;\mathrm{direction}\;\mathrm{dynamic}\;\mathrm{simulation}\;\text{-}\left(\max\left(\tau_{1},\tau_{2},\tau_{3},\tau_{4},\tau_{5},\tau_{6}\right)\right)\right\}$$

The pseudocode for optimizing the overall structural parameters is shown in Algorithm 1, and Figure 7 shows the flowchart of the GA.
**Algorithm 1** Pseudocode for parameter optimization of the 6-DOF robot.**Require:** The maximum number of iterations *M***Ensure:** Fitness function = $C-f_{op}$ (where *C* is selected to ensure that the fitness value is not negative)  1:Initialize the population size to *N*  2:Initialize the evolutionary iteration counter t=0  3:Set the crossover probability $P_{c}$ and the mutation probability $P_{m}$  4:Randomly generate individuals as the initial population *P*  5:**while** t<M **do**  6:   Calculate the fitness of each individual in the population *P*  7:   Select excellent individuals based on fitness  8:   **if** Adaptability meets the conditions **then**  9:     **return** The individuals with the best fitness obtained during the evolution process     are output as the optimal solution10:   **else**11:     Select excellent individuals based on fitness12:     Generate new solutions through crossover operations. Increase the population     diversity through mutation operations13:     The selection, crossover, and mutation operations yield the next     generation population P=P+1, t=t+114:     **return** to step 515:   **end if**16:**end while**17:**return** The individuals with the best fitness obtained during the evolution process are output as the optimal solution

The simulation is designed such that the robot platform performs uniformly accelerating linear motion in the *Z*-direction; the acceleration is 5 m/s^2^ (exceeding that of the human foot during walking) and the simulation time is 0.2 s. Figure 8 shows the process for the brief simulation. The initial values of the structural parameters are determined after several trials to be l=L=1000 mm; D=800 mm; d=447 mm; and d=75 mm. The optimization objective $f_{op}$ is used to determine the minimum joint driving force, where the larger the fitness in the genetic algorithm is, the better the individual is. Therefore, $Fitness\_function=C-f_{op}$ is selected to evaluate the performance associated with the mechanistic parameters. The *C* is a parameter that ensures the Fitness_function is positive. During each iteration of the genetic algorithm, each individual is dynamically simulated to obtain fitness values. For the 3-RPS robot, the design variables ${\left[R,r,H\right]}^{T}$ constitute a 3-dimensional design space. The same optimization method and objectives are used as for the 6-SSP robot.

Under the initial parameters of the 6-SSP robot, the optimization objective function value, which represents the maximum simulated joint torque, was approximately 3500 N. The optimization process aimed to reduce this torque by adjusting the robot’s structural parameters, thereby alleviating the burden on the robot’s power source and enhancing overall efficiency and stability. After approximately 10 iterations, the objective function value gradually decreased and stabilized, indicating the effectiveness of the optimization process. The maximum simulated joint torque decreased from 3500 N to 1500 N, a reduction of about 57%. At this point, the optimization process converged, and the objective function value stabilized, indicating that the optimal solution had been achieved. Ultimately, the robot attained the optimal structural parameters, which better support the robot’s performance in various motion modes. The optimized structure effectively reduced the joint torque and improved overall stability and performance. The final optimal structural parameters are listed in Table 2.

The 6-SSP robot is dynamically simulated using both the initial and optimal structural parameters.The schematic representation of the simulation process is presented in Figure 8. Figure 9a,b shows the driving forces for the six joints during the simulation, as determined by the initial and optimized parameters, respectively. The optimized structure significantly enhances the simulated motion in a specific direction. Optimization results in a reduction in the length of the rods, which aids in overcoming gravitational forces. The optimal structural parameters of the 6-SSP robot notably reduce the joint driving force during motion in the direction of gravity, thereby significantly improving the robot’s mechanical performance in that direction.

## 4. Dynamic Simulation and Analysis

The 6-SSP device is capable of performing global positioning; however, its parallel configuration limits its rotational capabilities. The motion and force transmission from the power source during rotation are relatively weak, resulting in a constrained rotational workspace and a high demand for joint driving forces. By integrating the 3-RPS and 6-SSP devices, redundant DOFs are introduced, offering multiple kinematic solutions. A careful allocation of these DOFs facilitates the optimization of the solution selection process. The 3-RPS device, which excels in rotational motion, is designated to execute the target rotational movements, while the 6-SSP device is solely responsible for linear motion. This strategic distribution of DOFs enhances motion performance by evenly distributing the load across the redundant devices, thereby reducing the burden on the 6-SSP device. Foot posture and position data during walking are collected from a volunteer wearing signal acquisition devices. The collected data are processed to address missing or anomalous values and undergo Kalman filtering, resulting in a self-constructed gait dataset, which serves as the trajectory for the motion platform. Dynamic simulations are then performed, with the robot’s motion pedals controlled to follow the predefined gait trajectory. The joint driving forces are subsequently monitored and analyzed. In the simulation environment, a total load of 100 kg is applied, with each of the two motion pedals supporting 50 kg. Two simulation groups are considered as follows: Group 1, where all movements are executed solely by the 6-SSP device; and Group 2, where movements are performed by the 9-DOF device. The motion DOFs are allocated as described above. Figure 10 illustrates the rationale behind the simulation verification experiment.

Figure 11a,b shows the gait data collected for the left and right feet of the volunteer, respectively. The figure shows that the movement process consists of four gait cycles, where the average and total duration of each gait cycle are 3.4 s and 13.6 s, respectively. The DOFs during walking mainly consist of movement in the *X*- and *Z*-directions and rotation around the *Y*-axis. When the volunteer is walking, the step length of a single foot is approximately 300 mm, the step height is approximately 150 mm, and the maximum pitch angle of the foot is approximately $20^{\circ }$.

Figure 12 shows the dynamic simulation results for Group 1, where all the gait movements are performed by the 6-SSP device alone. Figure 12a shows the driving forces on each linear active pair of the left device. Figure 12b shows the driving forces on each linear active pair on the right device.

Figure 13 shows the dynamic simulation results for Group 2, where all actions are performed by the 9-DOF device. Figure 13a,b shows the driving forces on each electric cylinder of the left and right 3-RPS device, respectively, and Figure 13c,d shows the driving forces on each prismatic active pair of the right and left 6-SSP device, respectively. In the two sets of simulations, the amplitude value of driving force (AVDC) and the Root Mean Square (RMS) of the motion pair in the 6-SSP device and 3-RPS device were, respectively, detected during the simulation process. The results are shown in Figure 14.

If the 6-SSP device performs all the movements alone during the simulation process, the maximum driving force exerted by the left 6-SSP device appears at LP5 at 4.92 s, with a peak value of 2855 N. At this time, the angular position of the left platform for motion around the Y-axis reaches a maximum of $20.3^{\circ }$. The maximum RMS of 993*N* occurs at LP4, indicating that LP4 at the left 6-SSP device needs to provide a large quantity of energy during motion. The maximum driving force exerted by the right 6-SSP device also occurs at LP5 at 6.15 s and a peak value of 2653 N. At this time, the angular position of the right platform for motion around the Y-axis reaches a maximum (also $20.3^{\circ }$). The maximum RMS of 1151 N occurs at LP5.

If the 6-SSP and 3-RPS devices work together, the 6-SSP device performs translational motion and the 3-RPS device performs rotational motion. During the simulation process, the maximum driving force exerted by the left 6-SSP device occurs at LP5 at 4.56 s, with a peak value of 1434 N. The maximum RMS of 678 N occurs at LP1. The maximum driving force exerted by the left 3-RPS device occurs at EC3 at 4.96 s, with a peak value of 584 N. The maximum driving force exerted by the right 6-SSP device also occurs at LP5 at 10.02 s, with a peak value of 1182 N. The maximum RMS of 683 N occurs at LP1. The maximum driving force exerted by the right 3-RPS device also occurs at EC3 at 6.14 s, with a peak value of 562 N. Table 3 summarizes the data obtained from the simulation experiment. Figure 15 shows the difference between the two simulation results.

## 5. Discussion

In this study, the structural parameters of the 6-SSP robot were optimized. The optimized structural parameters resulted in a significant reduction of approximately 57% in the joint driving torque required to overcome gravity during motion in the simulation environment. This optimization effectively alleviated the burden on the robot’s power source, significantly reducing its pressure. The rational structural parameters enhanced the mechanical performance of the 6-SSP robot, making task execution easier and improving the robot’s operational capabilities and efficiency.

Furthermore, by introducing the 3-RPS mechanism, the load and motion distribution of the 6-SSP robot were further optimized. The introduction of redundant DOFs prevented the 6-SSP mechanism from performing rotational movements, which it is less proficient at, allowing the robot to focus more on simulating and replicating the linear components of rehabilitation movements. When performing human-like walking gaits, compared to the 6-SSP mechanism alone, the 9-DOF robot demonstrated a reduction of approximately 40% in the RMS of the joint driving torque and a 50% decrease in AVDC, significantly improving both the robot’s energy consumption and load-bearing pressure. At the same time, the mechanical performance of the 6-SSP mechanism was enhanced, enabling the robot to perform more complex and diverse rehabilitation movements. This provides a solid foundation for the development of the robot’s functionalities and the design of its control systems.

## 6. Conclusions

This study analyzes the mechanical performance of a 9-DOF redundant rehabilitation robot, with the following key contributions:

First, the robot’s structural parameters are optimized by selecting key design variables, including joint configuration, linkage shape, and dimensions. A GA is used to optimize these parameters, aiming to enhance mechanical performance and reduce joint driving force. The optimization minimizes the maximum joint driving force during simulated motion, determined as the fitness function. This process yields an optimal parameter set that improves mechanical performance and reduces joint force during specific movements.

Second, integrating a 3-RPS redundant device with a 6-DOF parallel device expands the robot’s workspace and flexibility while enhancing the mechanical properties. A custom gait dataset, including foot posture and position data, is collected from volunteers and used as the motion trajectory for the robot’s pedals. A dynamic simulation controls the pedals to follow the gait trajectory, monitoring joint force. The results show that the 3-RPS device reduces both joint driving force and energy consumption for the 6-SSP device’s motion.

However, certain limitations exist in this research, particularly in the analysis and exploration of the robot’s mechanical performance under additional operational modes. As part of ongoing research, future work will build upon these findings to investigate the motion control and experimental validation of the rehabilitation robot, incorporating various movement modes such as stair climbing. This will further expand the robot’s capabilities, offering a more stable and effective therapeutic tool for patients with balance disorders.

## Figures and Tables

**Figure 1 biomimetics-10-00199-f001:**
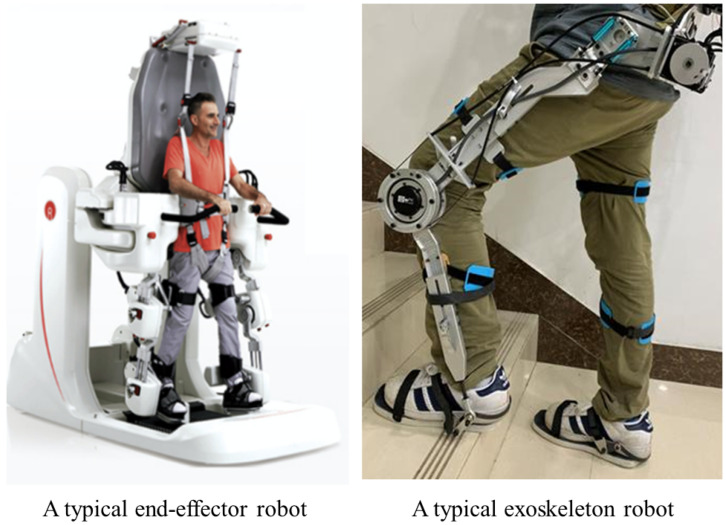
Typical robots for the rehabilitation training of lower limbs [4,5].

**Figure 2 biomimetics-10-00199-f002:**
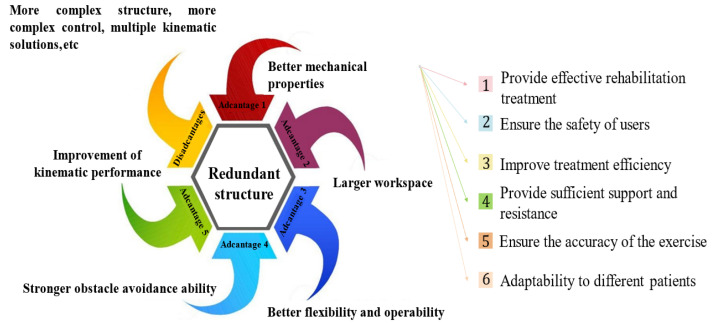
Importance of the mechanical properties of rehabilitation robots.

**Figure 3 biomimetics-10-00199-f003:**
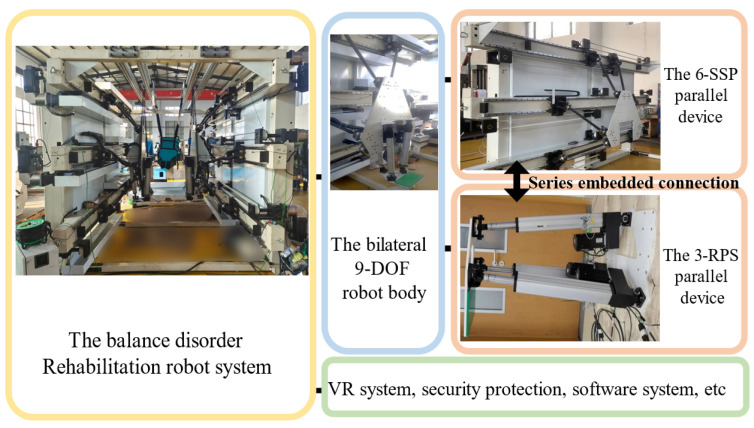
Components of the rehabilitation training system.

**Figure 4 biomimetics-10-00199-f004:**
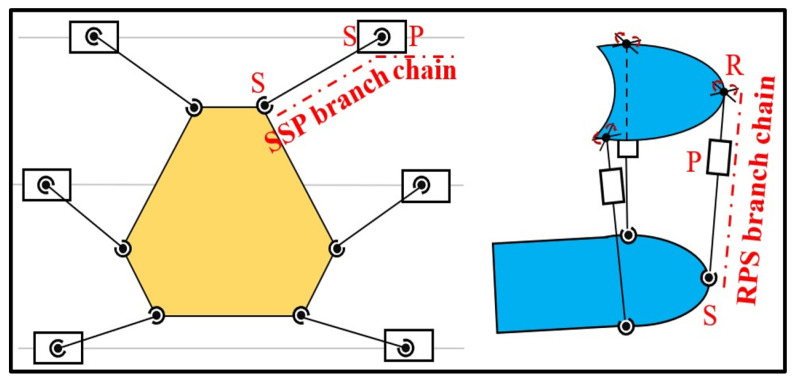
Structural diagrams of the 6-SSP and the 3-RPS devices.

**Figure 5 biomimetics-10-00199-f005:**
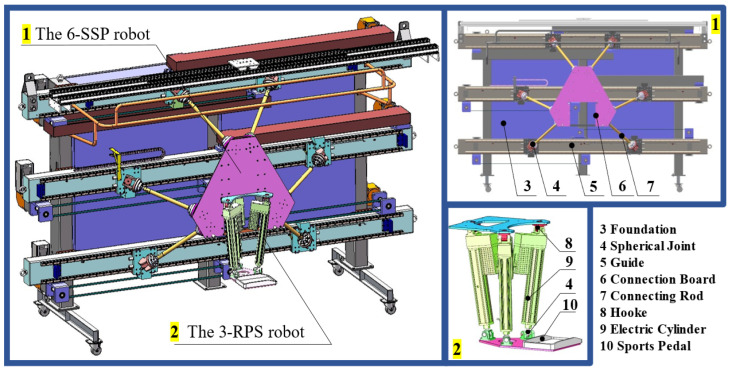
Structural compositions of the two devices.

**Figure 6 biomimetics-10-00199-f006:**
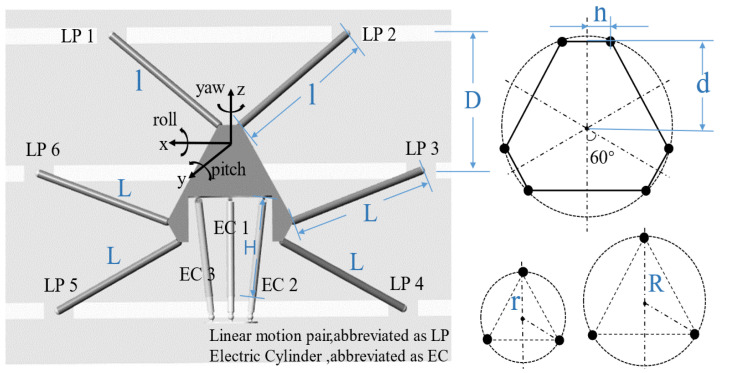
Key structural dimensions and joint definitions.

**Figure 7 biomimetics-10-00199-f007:**
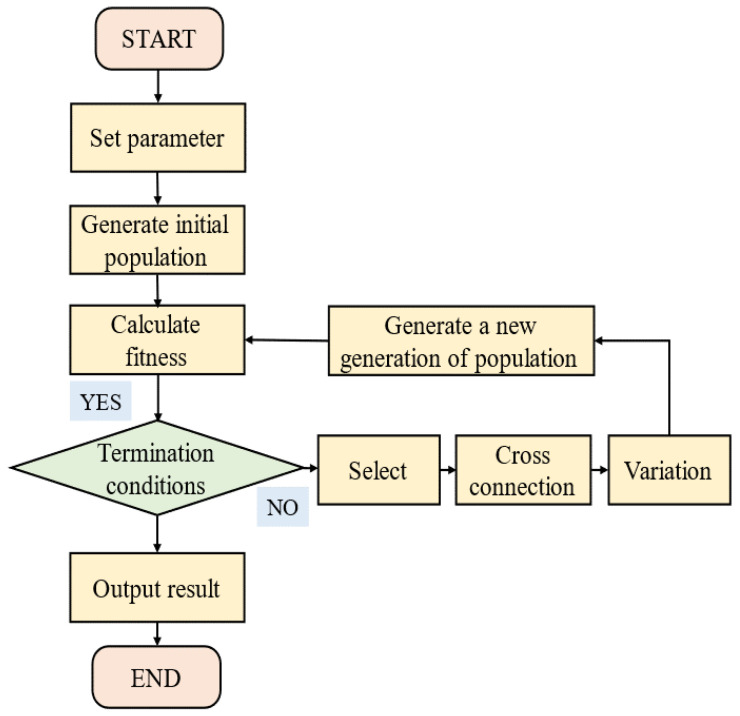
Genetic algorithm flow chart.

**Figure 8 biomimetics-10-00199-f008:**
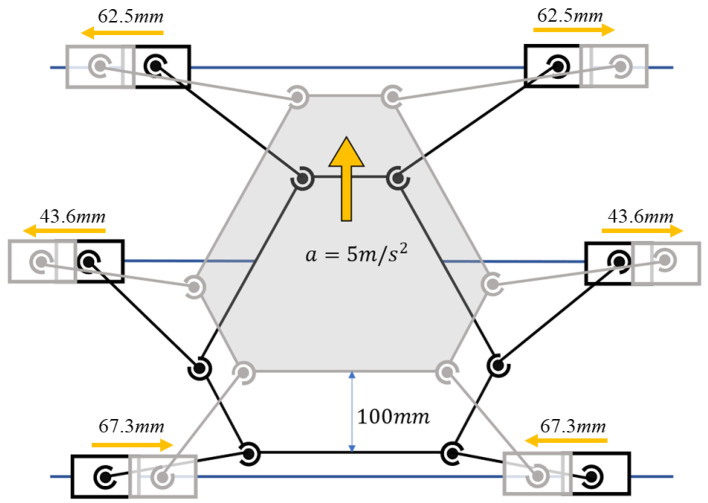
Sketch map of the simulation process.

**Figure 9 biomimetics-10-00199-f009:**
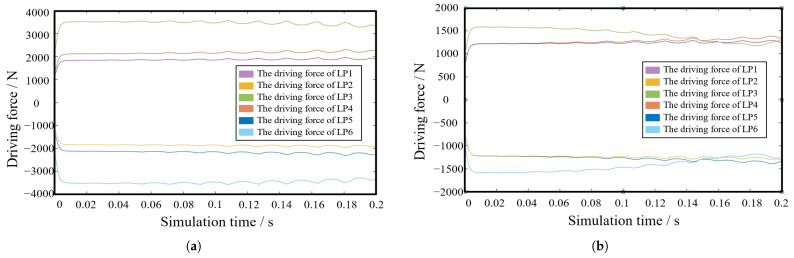
Dynamics simulation results. (**a**) Joint driving force when simulated under the given initial parameters’ motion. (**b**) Under the optimal parameters.

**Figure 10 biomimetics-10-00199-f010:**
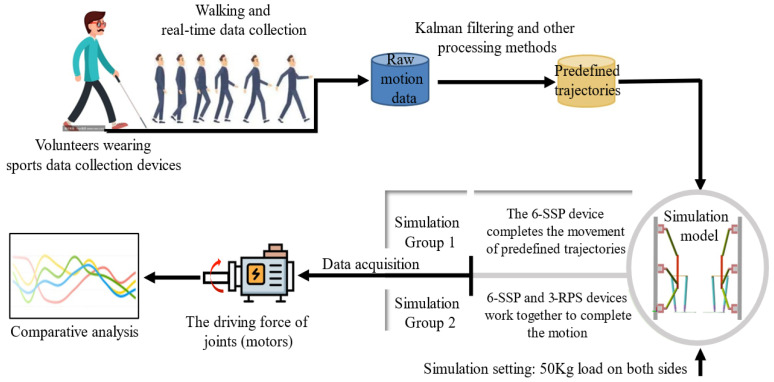
Rationale underlying the simulation verification experiment.

**Figure 11 biomimetics-10-00199-f011:**
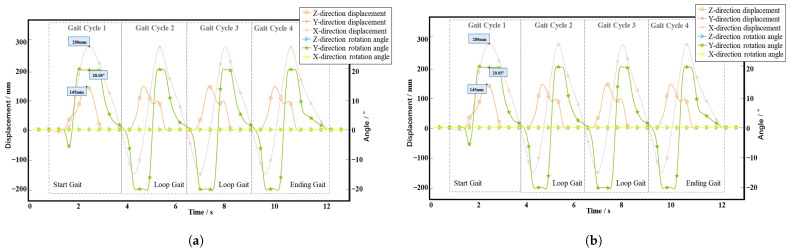
Gait data collected from a walking volunteer. (**a**) Data collected from the left foot. (**b**) Data collected from the right foot.

**Figure 12 biomimetics-10-00199-f012:**
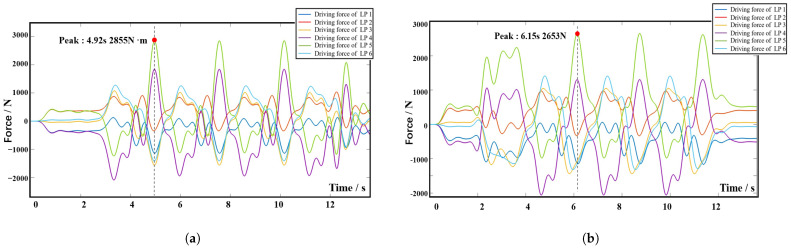
Simulation results for Group 1. (**a**)The left 6-SSP device. (**b**) The right 6-SSP device.

**Figure 13 biomimetics-10-00199-f013:**
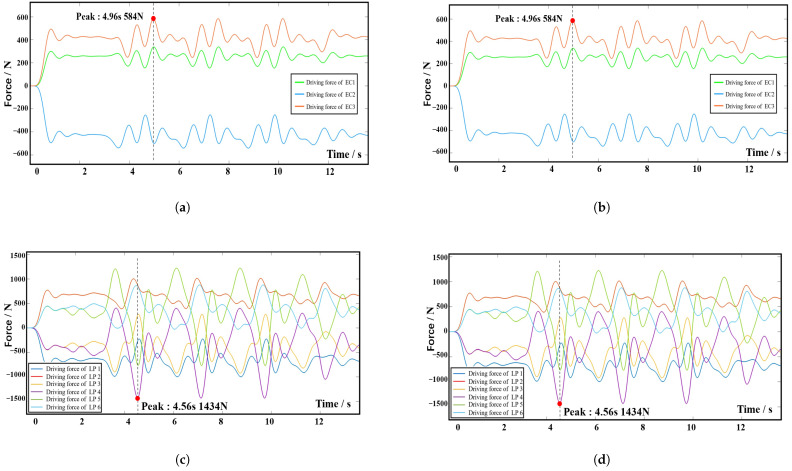
Simulation results for Group 2. (**a**) The left 3-RPS device. (**b**) The right 3-RPS device. (**c**) The left 6-SSP device. (**d**) The right 6-SSP device.

**Figure 14 biomimetics-10-00199-f014:**
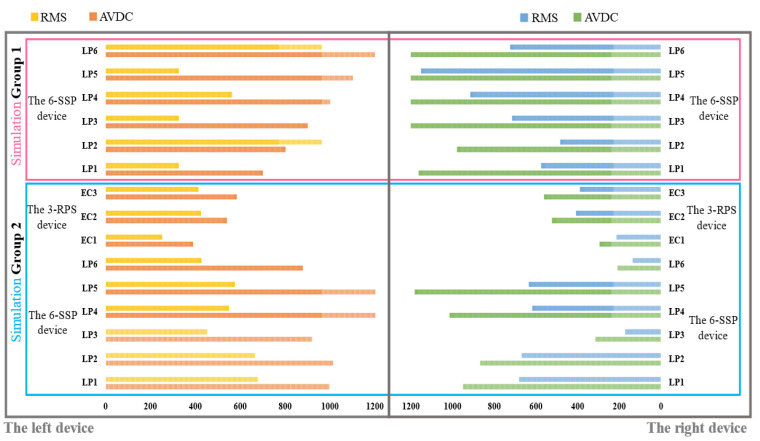
AVDC and MRS of the devices.

**Figure 15 biomimetics-10-00199-f015:**
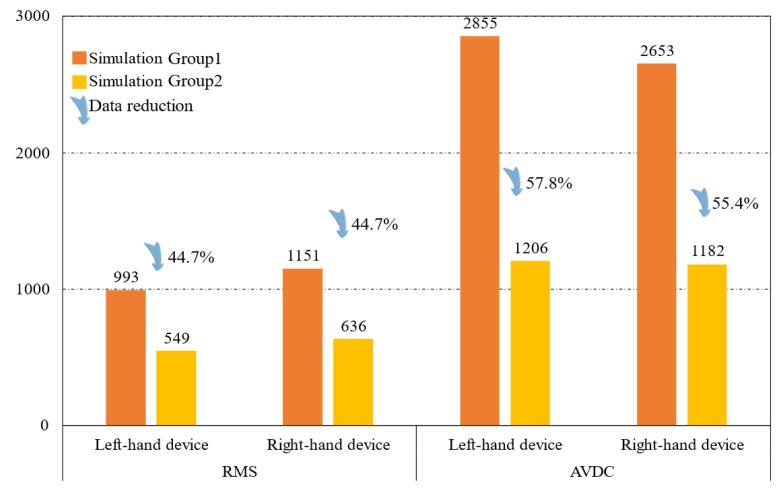
Comparative analysis of the two simulation results.

**Table 1 biomimetics-10-00199-t001:** Principle of DOF allocation.

Device	DOF	X	Y	Z	Roll	Pitch	Yaw	Total
The 6-SSP device	Having	✔	✔	✔	✔	✔	✔	6
	Using	✔	✔	✔			✔	4
The 3-RPS device	Having			✔	✔	✔		3
	Using			✔	✔	✔		3
The 9-DOF device	Using	✔	✔	✔	✔	✔	✔	6

**Table 2 biomimetics-10-00199-t002:** Key structural parameters of the robot.

Device	Parameter	Symbol	Value
6-SSP device	Length of the connecting rod	*L*	944 mm
		*l*	796 mm
	Geometric shape of the connecting plate	*h*	76 mm
		*d*	466 mm
	Guide-rail spacing	*D*	863 mm
3-RPS device	Initial length of the electric cylinder	*H*	600 mm
	Hooke hinge distribution	*r*	250 mm
	Ball-joint distribution	*R*	350 mm

**Table 3 biomimetics-10-00199-t003:** Data obtained from the simulation experiment.

Device	LP	RMS		AVDC	
		Group1	Group2	Group1	Group2
**The left device**	LP1	451	678	1273	996
	LP2	508	666	1036	1014
	LP3	634	453	1563	920
	LP4	**993 ***	549	2091	1434
	LP5	934	575	**2855 ***	1206
	LP6	681	426	1411	878
**The right device**	LP1	577	683	1164	951
	LP2	483	671	980	869
	LP3	715	172	1437	316
	LP4	917	618	2060	1014
	LP5	**1151 ***	636	**2653 ***	1182
	LP6	725	517	1415	209

Data with * represent the maximum RMS or AVDC of the left or right device.

## Data Availability

The data used to support the findings of this study are available from the corresponding author upon request.

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
