# Peer review of "Optimization of Structural Parameters and Mechanical Performance Analysis of a Novel Redundant Actuation Rehabilitation Training Robot"

_biomimetics, 2025, doi:10.3390/biomimetics10040199_

Round 1
Reviewer 1 Report
Comments and Suggestions for Authors
In this paper, mechanical performance of a 9-DOF redundant robot is presented and analyzed. In this paper the following points should be considered:
1) What is the novelty of the presented robot ? ( This robot was presented in the previous work , [ ref. 9 ] ;
2) What is the relationship between the presented work to rehabilitation process ( as far as , it indicated in the paper title ) ?
3) The optimization process is not clear and the obtained results can not be evaluated ;
4) As far as , the robot was built , it is recommended that the simulation results be compared with experimental results.
The English text should be improved .
Reviewer 2 Report
Comments and Suggestions for Authors This paper is focused on that the mechanical performance of a 9-DOF series-parallel hybrid
redundant device was evaluated for its application as a rehabilitation training robot for
patients with balance disorders. The redundant structural design enhances the robot’s
movement flexibility, optimizes load distribution, and mitigates stress concentration at
local joints or links. A genetic algorithm (GA) was used to optimize the robot’s overall
structural parameters, with the aim of reducing joint driving forces.
Figure 1 should be referenced because of literature work.
Where is the idea of The importance of mechanical properties of rehabilitation robots.
Kinematic and dynamic parameters of the 6 SSP and 3RSP of the mechanical systems should be given with a table.
Stroke of each piston should be given for the universal structure of the system on Figure 8.
What are the parameters of the arrows on Figure 10.
It is not clear to understand the system performance with wariation of he forces (see on Figures 9 (a) and (b)).
What is the reason to choose 9-DOF redundant rehabilitation robot. It is important to choose 8 DOF or 10 DOF redundant rehabilitation robot for walking stability and good position.
What is the reason for big differeneces between RMS errors for each LP1-LP6 . It should be similar variations. The papershuld be improved with good discussion section.
Comments on the Quality of English LanguageThe english should be improved with good technical sound.
Reviewer 3 Report
Comments and Suggestions for Authors
Dear Authors, please find my comments, questions, and suggestions below.
You should add the names of the robot types in Figure 1 to make it more clear to readers.
The Introduction section should be expanded. You should clearly state the main goal and objectives of the study, the novelty, results, and significance of findings. Please add this.
In section 2.2, you described the principle of DOF decomposition and motion allocation. Is this the only way to guarantees the uniqueness of the kinematic solution? If not, could you please provide more information about how you arrived at this decision?
As you know, genetic algorithms tend to converge to local optima rather than the global optimum in many problems. Are you sure that the genetic algorithm you used was the best possible solution? Have you considered using other optimization algorithms such as simulated annealing? You could write about it briefly in the beginning of section 3. It would be interesting for some readers.
The difference between the two graphs in Figure 9 is difficult to see. In my opinion, you should add a table with numerical values at key points or providing an estimate of the difference in the text of the article.
I did not find any references to Figures 13 and 14 in the text. Please add them. Also, please discuss properly the diagrams in Figure 14 and add this information to the article. There are also typos in Figure 14. The caption for Figure 14 says "MRS" and the legend says "MSR". I suppose there should be "RMS" here. Please correct it.
In general, I think your article deserves publication in Biomimetics Journal after major revision.
Round 2
Reviewer 1 Report
Comments and Suggestions for Authors
The revised article is improved about the proposed idea , but the previous main points and comments are still exist in revised paper.
Comments on the Quality of English LanguageThe English text should be improved for readability.
Reviewer 2 Report
Comments and Suggestions for Authors
The paper should be improved with discussion of comparison.
Comments on the Quality of English LanguageEnglish should be improved with technical sound for readers.
Reviewer 3 Report
Comments and Suggestions for Authors
Dear Authors,
thanks for improving the manuscript according to my comments and thanks for detailed response to my suggestions. I have no more questions for you. I think now your article can be published in the Biomimetics Journal.
Author Response
We are very glad to receive your recognition! Thank you once again for your careful review and guidance!